# Factors associated with perinatal mortality in sub-Saharan Africa: A multilevel analysis

**Meklit Melaku Bezie**[1]\*, **Hiwot Altaye Asebe**[2], **Angwach Abrham Asnake**[3], **Bezawit Melak Fente**[4], **Yohannes Mekuria Negussie**[5], **Zufan Alamrie Asmare**[6], **Mamaru Melkam**[7], **Beminate Lemma Seifu**[2]

1 Department of Public Health, Institute of Public Health, College of Medicine and Health Sciences and Comprehensive Specialized Hospital, University of Gondar, Gondar, Ethiopia, 2 Department of Public Health, College of Medicine and Health Sciences, Samara University, Samara, Ethiopia, 3 Department of Epidemiology and Biostatistics, School of Public Health, College of Medicine and Health Sciences, Wolaita Sodo University, Wolaita Sodo, Ethiopia, 4 Department of General Midwifery, School of Midwifery, College of Medicine & Health Sciences, University of Gondar, Gondar, Ethiopia, 5 Department of Medicine, Adama General Hospital and Medical College, Adama, Ethiopia, 6 Department of Ophthalmology, School of Medicine and Health Science, Debre Tabor University, Debre Tabor, Ethiopia, 7 Department of Psychiatry, University of Gondar College of Medicine and Health Science, Gondar, Ethiopia

\* mesiyemaki@gmail.com

**Data Availability Statement:** The data utilized in this study were sourced from the Demographic and Health Surveys (DHS) program and are publicly available through the DHS program's

## Abstract

### Background

Perinatal mortality is a major global public health concern, especially in sub-Saharan Africa (SSA). Despite perinatal mortality being a major public health concern in SSA, there are very limited studies on the incidence and factors associated with perinatal mortality. Therefore, we aimed to investigate the factors associated with perinatal mortality in SSA.

### Methods

A secondary data analysis was conducted based on the Demographic and Health Survey (DHS) data of 27 SSA countries. About 314,099 births in the preceding five years of the surveys were considered for the analysis. A multilevel binary logistic regression model was fitted to identify factors associated with perinatal mortality. Deviance (-2Log-Likelihood Ratio (LLR)) was used for model comparison. The Adjusted Odds Ratio (AOR) with the 5% Confidence Interval (CI) of the best-fitted model was used to verify the significant association between factors and perinatal mortality.

### Results

The perinatal mortality rate in sub-Saharan Africa (SSA) was 37.31 per 1,000 births (95% CI: 36.65, 37.98). In the final best-fit model, factors significantly associated with higher perinatal mortality included media exposure (AOR: 1.12, 95% CI: 1.08, 1.17), maternal age $\geq$ 35 years (AOR: 1.13, 95% CI: 1.06, 1.21), health facility delivery (AOR: 1.10, 95% CI: 1.06, 1.15), having 2–4 births (AOR: 1.35, 95% CI: 1.25, 1.47), five or more births (AOR: 1.69, 95% CI: 1.53, 1.86), residence in West Africa (AOR: 1.30, 95% CI: 1.24, 1.36) or Central Africa (AOR: 1.05, 95% CI: 1.00, 1.11), rural residency (AOR: 1.08, 95% CI: 1.02, 1.13), and

website. Access to these data requires registration and can be obtained at https://dhsprogram.com/data/dataset_admin/login_main.cfm.

**Funding:** The author(s) received no specific funding for this work.

**Competing interests:** The authors have declared that no competing interests exist.

**Abbreviations:** ANC, Antenatal Care; AOR, Adjusted Odds Ratio; CI, Confidence interval; DHS, Demographic and Health Survey; EA, Enumeration Area; PSU, Primary Sampling Unit; WHO, World Health Organization; SSA, Sub-Saharan Africa.

difficulty accessing a health facility (AOR: 1.06, 95% CI: 1.02, 1.10). In contrast, factors significantly associated with lower perinatal mortality were a preceding birth interval of 2–4 years (AOR: 0.70, 95% CI: 0.67, 0.74) or five or more years (AOR: 0.91, 95% CI: 0.84, 0.97), Antenatal Care (ANC) visit (AOR: 0.66, 95% CI: 0.63, 0.69), higher education levels (AOR: 0.82, 95% CI: 0.73, 0.93), middle household wealth (AOR: 0.93, 95% CI: 0.88, 0.98), and richer household wealth (AOR: 0.93, 95% CI: 0.87, 0.99).

## Conclusion

Perinatal mortality was a major public health problem in SSA. Maternal socio-demographic, obstetrical, and healthcare-related factors are significantly associated with perinatal mortality. The findings of this study highlighted the need for holistic healthcare interventions targeting enhancing maternal healthcare services to reduce the incidence of perinatal mortality.

## Background

Perinatal mortality is defined as the death of a fetus after 28 weeks of gestation (stillbirth) plus the death of a neonate within 7 days of life [1]. Annually, over 6.9 million perinatal deaths occur globally [2–5], and almost all cases occur in Low-income and Middle-income Countries (LMICs) [5]. Despite the substantial reduction in global perinatal mortality [6], LMICs such as sub-Saharan African (SSA) countries continue sharing the huge burden of global perinatal deaths [7]. Annually, nearly 7 million perinatal deaths occur globally, including 3 to 4 million stillbirths and 3 million early neonatal deaths, with approximately 99% of these deaths taking place in low- and middle-income regions, predominantly in sub-Saharan Africa [8].

LMICs are still seemingly walking a patchy road towards achieving the Sustainable Development Goal (SDG), which is to end preventable deaths of under-five children in 2023 [9]. Of the global 6.9 million perinatal deaths, the vast majority (98%) of these deaths were reported to have occurred in SSA [10]. Over three-quarters of perinatal mortality is due to preventable causes [11,12]. The perinatal mortality rate indicates the accessibility and quality of healthcare for the mother and baby [13]. For millions of families in low- and middle-income nations, perinatal mortality continues to be one of the most devastating pregnancy outcomes [14]. Currently, available evidence shows that perinatal mortality reflects the quality of maternal healthcare services during pregnancy, delivery, and postnatal care [15,16]. It results from the complex interaction of individual-level factors linked to maternal lifestyle and obstetric complications, which could be exacerbated by underlying community-level factors such as poor access to quality maternal and newborn health services.

Perinatal mortality is a key indicator of women's health-seeking behavior, maternal healthcare access, socioeconomic status, and the standard of obstetric care [17]. Beyond losing the child, perinatal mortality causes emotional and psychological stress for the mother. Mother may also experience social stigma or strained relationships with their partners [18]. In addition, it has a significant financial cost, worsens the health of the mother, perpetuates the cycle of poverty, and impedes social and economic development, especially in Sub-Saharan Africa (SSA), where child mortality is eleven times higher than in developed nations [19].

Previous studies published on determinants of perinatal mortality revealed that maternal education [20], parity [21], residence [22], preceding birth interval [23], place of delivery [24], number of Antenatal Care (ANC) [22], media exposure [25] and wealth status were

significantly associated with perinatal mortality. Several interventions to lower the Perinatal Mortality Rate (PMR) have been implemented globally, aiming to reduce stillbirths and neonatal deaths to 12 per 1,000 births by 2030 through the adoption of the Every Newborn Action Plan (ENAP) [12,26,27]. Launched in 2014 by the World Health Organization (WHO) and the United Nations Children's Fund (UNICEF), ENAP seeks to prevent newborn deaths and stillbirths while improving maternal and child health worldwide. However, PMR in sub-Saharan Africa remains alarmingly high compared to developed nations [28].

Despite perinatal mortality continuing to be a major public health problem in SSA, there are limited published studies on the magnitude and factors associated with perinatal mortality in SSA. Therefore, we aimed to investigate the magnitude and factors associated with perinatal mortality in SSA based on Demographic and Health Survey (DHS) data from 27 SSA countries. This study will guide program planners and decision-makers in designing evidence-based interventions to reduce perinatal mortality in SSA.

## Methods

### Study design and sample

We analyzed DHS data from 27 sub-Saharan African countries. The DHS data is collected every five years to update basic health and health-related indicators of the country. It is assumed to be nationally representative, where a multi-stage stratified sampling is employed to recruit representative samples. The Enumeration Area (EA) served as the primary sampling unit, while households were the secondary sampling unit. Data on perinatal mortality and independent variables were extracted from the Births Record (BR) dataset. Key variables from 27 sub-Saharan African (SSA) countries were collected and combined for the final analysis. The dataset included a total of 314,099 births from women of reproductive age within the five years preceding the survey.

### Dependent variable

The dependent variable for this study was perinatal mortality. Perinatal mortality was a composite of stillbirth and early neonatal death (deaths occurred before 7 days of birth) [29,30]. The outcome variable (Perinatal Mortality) was binary and recoded as "Yes" and "No".

### Covariates

We used both individual- and community-level variables for our study. Maternal Age (grouped as 15–24, 25–34, and 35–49 years), Media Exposure (grouped as No and Yes), Household Wealth Status (grouped as Poorest, Poorer, Middle, Richer, and Richest), Preceding Birth Interval (grouped as < 2 years, 2–4 years, ≥5 years and Primiparous), Place of Delivery (grouped as Home and Health Facility), Parity (grouped as One, Two–Four and Five births or above), ANC visits (grouped as No and Yes), Maternal Educational Status (grouped as No, Primary, Secondary and Higher Education), and Maternal Working Status (grouped as No and Yes) were the individual-level variables. Whereas, Residence (grouped as Urban and Rural), Distance from a Health Facility (grouped as a big problem and not a big problem), and SSA region (grouped as central, eastern, western, and southern regions) were considered as community-level variables.

### Data management and analysis

To ensure the study results were representative, we used DHS sample weights to account for unequal probability sampling across strata and non-response. Descriptive characteristics

including frequency and proportion were reported based on the weighted data to describe the study population. A multilevel binary logistic regression model was fitted to account for both the individual- and community-level variables. Stata version 17 was used for data management and analysis.

A multilevel binary logistic regression model was employed to account for the hierarchical structure of DHS data. The Intra-cluster Correlation Coefficient (ICC), Median Odds Ratio (MOR), and Likelihood Ratio (LR) tests were calculated to evaluate the variability of perinatal mortality across clusters. Four models were fitted, and model comparison was conducted using deviance (-2 Log-likelihood Ratio (-2LLR)), with the lowest value indicating the best-fitting model for the data. Deviance, expressed as the -2 Log-likelihood Ratio (-2LLR), is a statistical metric used to compare nested models in logistic regression and other likelihood-based approaches [31]. For nested models, it assesses whether the more complex model provides a better fit to the data than a simpler model by determining if the additional parameters in the complex model significantly enhance its performance. When it approaches zero, it indicates that the model fits the data exceptionally well. Model I was the null model fitted without covariates, Model II was the model with individual-level variables, Model 3 was the model with community-level variables and Model IV was the model with both individual and community-level variables. Variables with a P-value $\leq 0.2$ in the bivariable analysis were selected for multivariable multilevel logistic regression analysis. In the final best-fitted model, the Adjusted Odds Ratio (AOR) with 95% Confidence Interval (CI) was used to establish the statistical significance and strength of the association.

### Ethical consideration

For this study, we have received an authorization letter from the Measure DHS program for using the data. DHS provides publicly available de-identified data, so ethical approval is not needed. DHS employs several approaches to de-identify the data to ensure the privacy and confidentiality of respondents, such as data anonymization and the removal of personal identifiers.

## Results

### Descriptive characteristics of study participants

A total weighted sample of 314,099 births in the last five years preceding the survey was included (Table 1). More than two-thirds (68%) of participants resided in rural areas. About 110,470 (35.17%) and 109,700 (35%) of the mothers didn't have formal education and primary education, respectively. Approximately half of the mothers (47.65%) were within the age range of 25–34 years. Nearly one-fourth (22.9%) and 16.8% of participants belonged to the poorest and richest households, respectively. Only 13.33% of mothers had a history of terminating a pregnancy (Table 2).

### The perinatal mortality rate in SSA

The perinatal mortality rate in SSA was 37.31 per 1000 births (95% CI: 36.65, 37.98). It has varied across countries ranging from the lowest in Mozambique (10.8 per 1000 births) to the highest in Cote D'Ivoire (68.1 per 1000 births) (Fig 1).

### Factors associated with perinatal mortality

**Random effect analysis results.** We examined whether the multilevel binary logistic regression model was significant over the single-level binary logistic regression model using

**Table 1. Number of study participants included by country.**

| Country | Weighted sample | Percentage (%) |
|---|---|---|
| Angola | 13,276 | 4.23 |
| Burkina Faso | 12,210 | 3.89 |
| Benin | 13,542 | 4.31 |
| Burundi | 13,522 | 4.31 |
| Cote d'Ivoire | 9,690 | 3.08 |
| Cameroon | 10,024 | 3.19 |
| Ethiopia | 10,941 | 3.48 |
| Gabon | 6,045 | 1.92 |
| Ghana | 8,533 | 2.72 |
| Gambia | 7,608 | 2.42 |
| Guinea | 7,828 | 2.49 |
| Kenya | 17,316 | 5.51 |
| Liberia | 5,224 | 1.66 |
| Lesotho | 3,112 | 0.99 |
| Madagascar | 12,253 | 3.90 |
| Malawi | 34,499 | 10.98 |
| Mozambique | 5,491 | 1.75 |
| Nigeria | 34,018 | 10.83 |
| Rwanda | 8,267 | 2.63 |
| Sierra Leone | 9,718 | 3.09 |
| Chad | 18,635 | 5.93 |
| Togo | 6,706 | 2.14 |
| Tanzania | 10,806 | 3.44 |
| Uganda | 15,185 | 4.83 |
| South Africa | 3,541 | 1.13 |
| Zambia | 9,750 | 3.10 |
| Zimbabwe | 6,360 | 2.02 |
| Overall | 314,099 | 100 |

the Likelihood Ratio (LR) test. The LR test was statistically significant (p<0.05), indicating that the multilevel binary logistic regression model was best fitted to the single-level binary logistic regression analysis. Model comparison was employed by using deviance since models are nested models. Four models (mode I: 102197.9, model II: 100932, model III: 101997.6, and model IV: 100780.1) were fitted and model IV value (100780.1) was selected as the best-fitted model to explain perinatal mortality in SSA since it had the lowest deviance value. The MOR was 1.24 showing that if a mother of a child moved from a cluster with a low risk of perinatal mortality to a cluster with a high risk, the odds of experiencing perinatal mortality increased by 1.24 times.

**Fixed effect analysis results.** In multivariable multilevel logistic regression analysis, individual-level variables such as maternal age, maternal education, parity, ANC, preceding birth interval, and place of delivery, and community-level variables such as residence, distance to the health facility, and sub-Saharan African region had a statistically significant association with perinatal mortality.

The odds of perinatal mortality of babies to mothers aged 35–49 years were 1.13 times (AOR = 1.13, 95% CI: 1.06, 1.21) higher compared to those born to mothers aged 15–24 years. The odds of experiencing perinatal mortality among mothers who had higher education were decreased by 18% (AOR = 0.82, 95% CI: 0.73, 0.93) compared to mothers who had no formal

**Table 2. Descriptive characteristics of women aged 15–49 in sub-Saharan African countries.**

| Variables | Weighted frequency | Percentage (%) |
|---|---|---|
| **Residence** | | |
| Rural | 213,371 | 67.93 |
| Urban | 100,728 | 32.07 |
| **Maternal age (in years)** | | |
| 15–24 | 91,416 | 29.10 |
| 25–34 | 149,661 | 47.65 |
| ≥35 | 73,021 | 23.25 |
| **Maternal education** | | |
| No education | 110,470 | 35.17 |
| Primary | 109,700 | 34.93 |
| Secondary | 79,648 | 25.36 |
| Higher | 14,281 | 4.55 |
| **Household wealth status** | | |
| Poorest | 71,982 | 22.92 |
| Poorer | 67,690 | 21.55 |
| Middle | 62,682 | 19.96 |
| Richer | 59,357 | 18.90 |
| Richest | 52,288 | 16.68 |
| **Media exposure** | | |
| No | 121,460 | 38.67 |
| Yes | 192,639 | 61.33 |
| **Parity** | | |
| One birth | 49,888 | 15.88 |
| Two to four births | 163,314 | 51.99 |
| Five births and above | 100,896 | 32.12 |
| **Perceived distance to health facility** | | |
| A big problem | 144,906 | 46.13 |
| Not a big problem | 169,193 | 53.87 |
| **Preceding birth interval** | | |
| Primiparous | 75,343 | 23.99 |
| < 2 years | 44,353 | 14.12 |
| 2–4 years | 156,572 | 49.85 |
| ≥ 5 years | 37,831 | 12.04 |
| **Maternal working status** | | |
| No | 113,721 | 36.21 |
| Yes | 200,378 | 63.79 |
| **Place of delivery** | | |
| Home | 118,609 | 37.76 |
| Health facility | 195,490 | 62.24 |
| **ANC** | | |
| No | 123,981 | 39.47 |
| Yes | 190,118 | 60.53 |

education. The odds of perinatal mortality among mothers who belonged to middle and richer household wealth were decreased by 7% (AOR = 0.93, 95% CI: 0.88, 0.98) and 7% (AOR = 0.93, 95% CI: 0.87, 0.99) compared to those belonged to the poorest household, respectively. Mothers who had no media exposure had 1.12 times (AOR = 1.12, 95% CI: 1.08,

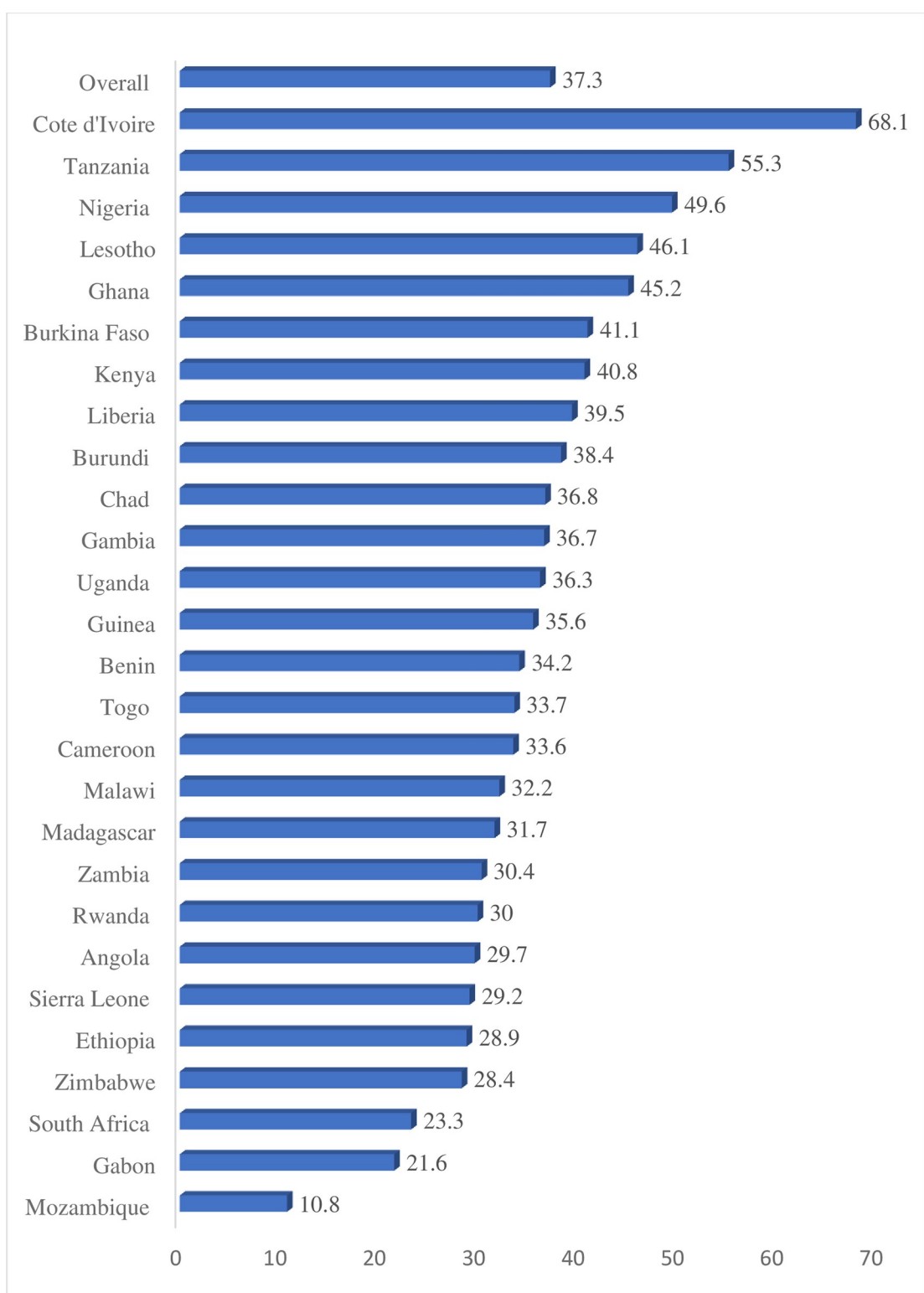

**Fig 1. Perinatal mortality rate across countries in SSA.**

1.17) higher odds of perinatal mortality compared to those who had media exposure. Multiparous and grand multiparous mothers had 1.35 times (AOR = 1.35, 95% CI: 1.25, 1.47) and 1.69 times (AOR = 1.69, 95% CI: 1.53, 1.86) higher odds of perinatal mortality compared to primiparous mother, respectively.

The odds of perinatal mortality among mothers who had preceding birth interval of 2–4 years and 5 years or above were decreased by 30% (AOR = 0.70, 95% CI: 0.67, 0.74) and 9% (AOR = 0.91, 95% CI: 0.84, 0.97) compared to those who had birth interval of less than 2 years, respectively. Mothers who gave birth at a health facility had 1.10 times (AOR = 1.10, 95% CI: 1.06, 1.15) higher odds of perinatal mortality than those who gave birth at home. The odds of perinatal mortality among mothers who had ANC visits were decreased by 34% (AOR = 0.66, 95% CI: 0.63, 0.69). Being a rural resident increased the odds of perinatal mortality by 8% (AOR = 1.08, 95% CI: 1.02, 1.13). Mothers in West Africa and Central Africa had 1.30 times (AOR = 1.30, 95% CI: 1.24, 1.36) and 1.05 times (AOR = 1.05, 95% CI: 1.00, 1.11) higher odds of perinatal mortality compared to mothers in East Africa, respectively. Mothers who had big problems accessing health care increased the odds of perinatal mortality by 6% (AOR = 1.06, 95% CI: 1.02, 1.10) (Table 3).

## Discussion

In our study, the perinatal mortality rate in SSA was found to be 37 per 1000 births which varied across regions. It was higher than a study reported in Indonesia [32] and India [33]. This difference may be due to the extensive socio-political and economic crises that sub-Saharan African countries have been experiencing, which are caused by both man-made and natural disasters that can contribute to increased perinatal mortality [34]. The findings of our study demonstrated that perinatal mortality in SSA remains a significant public health problem. The higher perinatal mortality rate is the result of a multiplicity of factors such as limited access to ANC, lack of quality of care during delivery, higher prevalence of poverty, low level of maternal education, maternal nutritional deficiency, teenage pregnancy, high prevalence of maternal infections (HIV, malaria, syphilis. . .etc), preterm births, low birth weight, and home delivery [35,36]. Additionally, in many African countries, entrenched cultural beliefs and practices affect maternal and neonatal care, such as the delayed initiation of breastfeeding, which can impact perinatal outcomes [37].

In the multilevel analysis variables such as media exposure, maternal age, parity, household wealth status, preceding birth interval, maternal education, place of delivery, ANC, distance to health facility, residence, and SSA region were significantly associated with perinatal mortality. The odds of perinatal mortality among mothers aged 35 years and above were higher compared to those aged 15–24 years. This finding is consistent with previous studies [38,39], likely due to the strong association between advanced maternal age and an elevated risk of adverse pregnancy conditions. These conditions include pregnancy-induced hypertension, antepartum hemorrhage, cesarean delivery, preterm delivery, low birth weight, and premature rupture of membranes, all of which can increase the risk of perinatal deaths [40,41]. Maternal education reduces the risk of perinatal mortality. This finding was in line with studies reported in Nigeria [42], Spain [43], and Pakistan [44]. It could be due to maternal education plays a significant role in improving maternal and perinatal health outcomes [45]. Educated mothers understand health-related information, are more likely to seek perinatal care, recognize danger signs of pregnancy, and take timely interventions [46]. In addition, educated mothers are highly likely to attend prenatal care early and regularly as well as more likely to give birth in healthcare facilities. Furthermore, mothers with higher education are empowered to make their own decisions about their health and childbirth.

**Table 3. Multilevel binary logistic regression analysis of factors associated with perinatal mortality in SSA.**

| Variables | Null model | Model II (AOR with 95% CI) | Model III (AOR with 95% CI) | Model IV (AOR with 95% CI) |
|---|---|---|---|---|
| **Maternal age (in years)** | | | | |
| 15–24 | | 1 | | 1 |
| 25–34 | | 0.97 (0.92, 1.02) | | 0.96 (0.91, 1.01) |
| ≥ 35 | | 1.15 (1.07, 1.23) | | 1.13 (1.06, 1.21) * |
| **Maternal education status** | | | | |
| No education | | 1 | | 1 |
| Primary | | 0.95 (0.91, 1.00) | | 1.05 (0.99, 1.10) |
| Secondary | | 0.89 (0.84, 0.95) | | 0.94 (0.89, 1.00) |
| Higher | | 0.79 (0.70, 0.89) | | 0.82 (0.73, 0.93) ** |
| **Household wealth status** | | | | |
| Poorest | | 1 | | 1 |
| Poorer | | 0.95 (0.90, 1.01) | | 0.94 (0.90, 1.00) |
| Middle | | 0.93 (0.88, 0.98) | | 0.93 (0.88, 0.98) ** |
| Richer | | 0.91 (0.86, 0.97) | | 0.93 (0.87, 0.99) ** |
| Richest | | 0.88 (0.82, 0.94) | | 0.92 (0.85, 1.00) |
| **Media exposure** | | | | |
| No | | 1.16 (1.11, 1.21) | | 1.12 (1.08, 1.17) * |
| Yes | | 1 | | 1 |
| **Parity** | | | | |
| < 2 | | 1 | | 1 |
| 2–4 | | 1.34 (1.24, 1.45) | | 1.35 (1.25, 1.47) * |
| ≥5 | | 1.64 (1.49, 1.81) | | 1.69 (1.53, 1.86) ** |
| **Preceding birth interval** | | | | |
| < 2 years | | 1 | | 1 |
| 2–4 years | | 0.71 (0.68, 0.75) | | 0.70 (0.67, 0.74) ** |
| ≥ 5 years | | 0.91 (0.85, 0.97) | | 0.91 (0.84, 0.97) * |
| Primiparous | | 1.20 (1.12, 1.30) | | 1.20 (1.12, 1.30) |
| **Place of delivery** | | | | |
| Health facility delivery | | 1 | | 1 |
| Home delivery | | 1.10 (1.05, 1.15) | | 1.10 (1.06, 1.15) * |
| **ANC visit** | | | | |
| No | | 1 | | 1 |
| Yes | | 0.66 (0.63, 0.69) | | 0.66 (0.63, 0.69) ** |
| **Residence** | | | | |
| Urban | | | 1 | 1 |
| Rural | | | 1.14 (1.09, 1.19) | 1.08 (1.02, 1.13) * |
| **SSA region** | | | | |
| East Africa | | | 1 | 1 |
| West Africa | | | 1.32 (1.27, 1.38) | 1.30 (1.24, 1.36) ** |
| Southern Africa | | | 1.02 (0.89, 1.17) | 1.11 (0.96, 1.27) |
| Central Africa | | | 1.10 (1.05, 1.16) | 1.05 (1.00, 1.11) |
| **Distance to health facility** | | | | |
| Not big problem | | | 1 | 1 |
| Big problem | | | 1.03 (0.99, 1.07) | 1.06 (1.02, 1.10) * |
| **Random effect analysis results** | | | | |
| LLR | -51098.94 | -50466.46 | -50998.78 | -50390.07 |

(*Continued*)

**Table 3.** (Continued)

| Variables | Null model | Model II (AOR with 95% CI) | Model III (AOR with 95% CI) | Model IV (AOR with 95% CI) |
|-----------|-----------|----------------------------|------------------------------|------------------------------|
| Deviance | 102197.9 | 100932.92 | 101997.6 | 100780.1 |

*ANC: Antenatal Care, AOR: Adjusted Odds Ratio, CI: Confidence Interval, LLR: Log-likelihood Ratio*

* $p < 0.05$

** $p < 0.01$.

The odds of perinatal mortality are high among mothers with five or more children compared to babies born to mothers who have fewer than two children. This study is in line with previous findings in Australia which stated that grand multiparous mothers are at risk of perinatal [47,48]. This may be due to multiparous mothers undergoing uterine and placental changes that increase the risk of adverse outcomes. Poor maternal nutrition can lead to low birth weight and higher stillbirth risk. Additionally, financial constraints in low-resource settings may limit access to quality health services for these mothers.

As the birth interval increases, it is significantly associated with lower perinatal deaths, which is consistent with studies in Bangladesh [49] and Ethiopia [23]. It could be due to having a longer preceding birth interval enable the mother to recover physically and psychologically, and helps their nutritional recovery which strengthens the mother's immunity system, which in turn helps to avoid maternal infection as well as that of the fetus [50]. ANC reduces the risk of perinatal mortality. This finding is consistent with the study findings in Ethiopia [51,52], and Gambia [53]. The possible explanation might be due to ANC makes it possible to track fetal development in order to provide timely interventions and preventive measures, as well as to diagnose pregnancy complications such as gestational diabetes and preeclampsia early on [54]. It creates the opportunity to connect mothers with health professionals, helping them prepare for birth and any emergencies [55]. Our study revealed that perinatal mortality was significantly higher in rural areas. This is consistent with the study done in India [56]. This could be due to the majority of rural mothers giving birth at home, so they are at higher risk of intrapartum and early postpartum complications [57].

Mothers who had big problems accessing healthcare had higher odds of perinatal mortality. This study is in line with findings in Ethiopia [58,59]. The rise in missed appointments and the postponed start of maternity care could serve as justification for this [60]. Furthermore, it can be challenging to promptly address complications like hemorrhage and obstructed labor in low socioeconomic households where transportation is a major challenge [61].

According to our findings, higher socio-economic class is associated with lower perinatal mortality. This is supported by a study done in the Netherlands [62], and in western China [63]. Our study states that babies born in health facilities had an increased risk of perinatal mortality as compared to babies born at home. This study supported the study done in Kenya [64]. This may be explained by the fact that babies born in health facilities often come from mothers who are already triaged as high-risk, inherently increasing the perinatal death rate compared to mothers with uncomplicated pregnancies who deliver at home. Additionally, usually due to transportation and other related constraints mothers in SSA mothers who arrive at health facilities [65,66], which in turn made them at increased risk of complications such as fetal distress, meconium aspiration, etc.

## Strengths and limitations of the study

The main strength of this study is the use used representative data from DHS of 27 sub-Saharan African countries, which involves a large sample size. This study also employed a

multilevel technique that considers the hierarchical nature of the survey and therefore, the results are reliable and robust. However, the data is cross-sectional, and the temporal relationship of causations cannot be established. This study might have a recall bias, for variables such as preceding birth, postnatal care, and antenatal care visits, which should be taken into account when interpreting our results.

## Conclusion

Perinatal mortality remains a significant public health concern in SSA. Advanced maternal age, high parity, being rural, having no media exposure, home delivery, and a big problem accessing health care were significantly associated with higher odds of perinatal mortality, while ANC, higher maternal education, belonging to a wealthier household, optimal and long preceding birth interval were significantly associated with lower odds of perinatal mortality. The findings of this study highlighted the need to enhance maternal education, ANC use, health facility delivery, and media exposure to reduce the incidence of perinatal mortality.

## Acknowledgments

We acknowledge the MEASURE DHS program for permitting the use of the dataset.

## Author Contributions

**Conceptualization:** Meklit Melaku Bezie, Hiwot Altaye Asebe, Angwach Abrham Asnake, Bezawit Melak Fente, Yohannes Mekuria Negussie, Zufan Alamrie Asmare, Mamaru Melkam, Beminate Lemma Seifu.

**Data curation:** Meklit Melaku Bezie, Hiwot Altaye Asebe, Angwach Abrham Asnake, Bezawit Melak Fente, Yohannes Mekuria Negussie, Zufan Alamrie Asmare, Mamaru Melkam, Beminate Lemma Seifu.

**Formal analysis:** Meklit Melaku Bezie, Hiwot Altaye Asebe, Angwach Abrham Asnake, Bezawit Melak Fente, Yohannes Mekuria Negussie, Zufan Alamrie Asmare, Mamaru Melkam, Beminate Lemma Seifu.

**Funding acquisition:** Meklit Melaku Bezie, Hiwot Altaye Asebe, Angwach Abrham Asnake, Bezawit Melak Fente, Yohannes Mekuria Negussie, Zufan Alamrie Asmare, Mamaru Melkam, Beminate Lemma Seifu.

**Investigation:** Meklit Melaku Bezie, Hiwot Altaye Asebe, Angwach Abrham Asnake, Bezawit Melak Fente, Yohannes Mekuria Negussie, Zufan Alamrie Asmare, Mamaru Melkam, Beminate Lemma Seifu.

**Methodology:** Meklit Melaku Bezie, Hiwot Altaye Asebe, Angwach Abrham Asnake, Bezawit Melak Fente, Yohannes Mekuria Negussie, Zufan Alamrie Asmare, Mamaru Melkam, Beminate Lemma Seifu.

**Project administration:** Meklit Melaku Bezie, Hiwot Altaye Asebe, Angwach Abrham Asnake, Bezawit Melak Fente, Yohannes Mekuria Negussie, Zufan Alamrie Asmare, Mamaru Melkam, Beminate Lemma Seifu.

**Resources:** Meklit Melaku Bezie, Hiwot Altaye Asebe, Angwach Abrham Asnake, Bezawit Melak Fente, Yohannes Mekuria Negussie, Zufan Alamrie Asmare, Mamaru Melkam, Beminate Lemma Seifu.

**Software:** Meklit Melaku Bezie.

**Supervision:** Meklit Melaku Bezie, Hiwot Altaye Asebe, Angwach Abrham Asnake, Bezawit Melak Fente, Yohannes Mekuria Negussie, Zufan Alamrie Asmare, Mamaru Melkam, Beminate Lemma Seifu.

**Validation:** Meklit Melaku Bezie, Hiwot Altaye Asebe, Angwach Abrham Asnake, Bezawit Melak Fente, Yohannes Mekuria Negussie, Zufan Alamrie Asmare, Mamaru Melkam, Beminate Lemma Seifu.

**Visualization:** Meklit Melaku Bezie, Hiwot Altaye Asebe, Angwach Abrham Asnake, Bezawit Melak Fente, Yohannes Mekuria Negussie, Zufan Alamrie Asmare, Mamaru Melkam, Beminate Lemma Seifu.

**Writing – original draft:** Meklit Melaku Bezie.

**Writing – review & editing:** Meklit Melaku Bezie, Hiwot Altaye Asebe, Angwach Abrham Asnake, Bezawit Melak Fente, Yohannes Mekuria Negussie, Zufan Alamrie Asmare, Mamaru Melkam, Beminate Lemma Seifu.

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
