## [Editor Report · Decision Letter 0]

24 Sep 2024

PONE-D-24-24034Factors associated with perinatal mortality in Sub-Saharan Africa: A multilevel analysisPLOS ONE

Dear Dr. Bezie,

Thank you for submitting your manuscript to PLOS ONE. After careful consideration, we feel that it has merit but does not fully meet PLOS ONE’s publication criteria as it currently stands. Therefore, we invite you to submit a revised version of the manuscript that addresses the points raised during the review process.

We look forward to receiving your revised manuscript.

Kind regards,

Alfredo Luis Fort, M.D., M.Sc., Ph.D.

Academic Editor

PLOS ONE

2. Please note that your Data Availability Statement is currently missing the repository name and/or the DOI/accession number of each dataset OR a direct link to access each database. If your manuscript is accepted for publication, you will be asked to provide these details on a very short timeline. We therefore suggest that you provide this information now, though we will not hold up the peer review process if you are unable.

Additional Editor Comments:

The study is of high importance and the methodology used is OK. However, there are a number of place where descriptions and sentence writing is not clear enough for a reader to understand, plus requiring some descriptions, etc. That's why it is sent back to the authors to ensure a person with high written knowledge of academic writing revises the manuscript entirely, and then resubmit to PLOS ONE. See a number of suggestions in the attached file. Thanks.

---

## [Author Response · Author response to Decision Letter 0]

10 Oct 2024

Author response

PLOS ONE Journal 

Manuscript title: Factors associated with perinatal mortality in Sub-Saharan Africa: A multilevel analysis

Manuscript ID: PONE-D-24-24034

Dear editor/reviewer. 

Dear all,

We would like to thank you for the constructive, building, and improvable comments on this manuscript that would improve the content of the manuscript. We considered each comment and clarification question of editors and reviewers on the manuscript thoroughly. Our point-by-point responses for each comment and question are described in detail on the following pages. Further, the details of changes were shown by track changes in the supplementary document attached.

Reviewer comments 

1. Abstract 

These variables will have to be put in special case, e.g., Media Exposure, or with inverted commas, e.g., "Media Exposure" (as it was in the report), so that it is clear to the reader...

Authors’ response: Thank you reviewer for the comments. We have accepted all the comments and revised them according to your suggestions. (See the Revised manuscript)

2. Background

- It would be good to put some figures here, so the reader can focus on differences between regions.

Authors’ response: Thank you reviewer for the suggestions. We have added the figures for the perinatal mortality rate. "Annually, nearly 7 million perinatal deaths occur globally, including 3 to 4 million stillbirths and 3 million early neonatal deaths, with approximately 99% of these deaths taking place in low- and middle-income regions, predominantly in sub-Saharan Africa". (See the revised manuscript)

- By which institution has this been advocated...WHO?...Important to instruct the reader...

Authors’ response: Thank you reviewer for the comment. We have included which institution advocated the ENAP. “Several interventions to lower the Perinatal Mortality Rate (PMR) have been implemented globally, aiming to reduce stillbirths and neonatal deaths to 12 per 1,000 births by 2030 through the adoption of the Every Newborn Action Plan (ENAP) [12, 26, 27]. Launched in 2014 by the World Health Organization (WHO) and the United Nations Children's Fund (UNICEF), ENAP seeks to prevent newborn deaths and stillbirths while improving maternal and child health worldwide. However, PMR in sub-Saharan Africa remains alarmingly high compared to developed nations”. (See the Revised manuscript). 

3. Methods 

- The sentence is unclear. Here it seems to mean that the birth record dataset was also used to extract data on "independent variables". Needs rephrasing.

Authors’ response: Thank you reviewer for the comments. We used the Births Record (BR) dataset. As you know there are several DHS datasets including the women, men, kids, births, couples, and household datasets, and the type of dataset to be used depends on the objective of the research. For the child and birth outcomes, we can use either the births record or kids record data. As per the DHS recode manual and guide to statistics, we used the BR dataset to extract the outcome and independent variables. "Data on perinatal mortality and independent variables were extracted from the Births Record (BR) dataset.”.

- Again, variables should be written differently from the sentence words, e.g., by capitalizing them, e.g., No, Yes, or better, by using inverted commas (e.g., "Yes", "No").

Authors’ response: Thank you reviewer for the comment. We have accepted the comment and made the revision. (See the Revised manuscript)

- "and deviance"? Is this an incomplete sentence? Please re-word/improve.

Authors’ response: Thank you for the comment. We have revised it. We used the deviance (-2LLR) for model comparison because the models were nested models where the lower the deviance value the better the model fits the data. A model with the lowest deviance value was chosen as the best-fitted model. We have added some details about deviance in the methods. "Four models were fitted, and model comparison was conducted using deviance (-2 Log-likelihood Ratio (-2LLR)), with the lowest value indicating the best-fitting model for the data. Deviance, expressed as the -2 Log-likelihood Ratio (-2LLR), is a statistical metric used to compare nested models in logistic regression and other likelihood-based approaches [31]. For nested models, it assesses whether the more complex model provides a better fit to the data than a simpler model by determining if the additional parameters in the complex model significantly enhance its performance. When it approaches zero, it indicates that the model fits the data exceptionally well”. (See the Revised manuscript)

- De-identified DHS data, with what?

Author's response: Thank you for the comment. We have revised the ethical consideration statement. In DHS, data was de-identified using data anonymization and removal of personal identifiers to ensure the participant's privacy and confidentiality. "For this study, we have received an authorization letter from the Measure DHS program for using the data. DHS provides publicly available de-identified data, so ethical approval is not needed. DHS employs several approaches to de-identify the data to ensure the privacy and confidentiality of respondents, such as data anonymization and the removal of personal identifiers”. (See the Revised manuscript)

4. Results

1. For the benefit of the reader, develop/explain this more, e.g., a way to generalize the residual sums of squares, to show the "goodness of fit"...etc.

Author's response: Thank you for raising your concern. For our study, we used LLR and deviance for model comparison. As you may know, deviance is -2 times the log-likelihood ratio of the model. Given our outcome variable is binary /categorical, the model fitness/adequacy is ideally assessed using LLR because the model is based on the maximum likelihood estimation, unlike the linear regression where least square regression is used. So, the residual sum of squares is the key metric for assessing the model fitness of linear regression models where the residuals are the difference between the predicted and observed values. Therefore, the residual sum of squares can not be used for the logistic regression model. So, for our study, we used the model fitness using the LR test and deviance. For further reading see https://stats.stackexchange.com/questions/237702/comparing-models-using-the-deviance-and-log-likelihood-ratio-tests. 

General comments

We have accepted all the editorial comments and revised them accordingly. (See the revised manuscript)

---

## [Decision Letter · Decision Letter 1]

6 Nov 2024

Factors associated with perinatal mortality in Sub-Saharan Africa: A multilevel analysis

PONE-D-24-24034R1

Dear Dr. Bezie,

We’re pleased to inform you that your manuscript has been judged scientifically suitable for publication and will be formally accepted for publication once it meets all outstanding technical requirements.

Kind regards,

Alfredo Luis Fort, M.D., M.Sc., Ph.D.

Academic Editor

PLOS ONE

Additional Editor Comments (optional):

Thank you for making the necessary adjustments and edits to improve your manuscript. It is now ready to be submitted to PLOS ONE for publication. Please see the attached file for one minor error to be corrected. Thanks.

**Comments to the Author**

1. If the authors have adequately addressed your comments raised in a previous round of review and you feel that this manuscript is now acceptable for publication, you may indicate that here to bypass the “Comments to the Author” section, enter your conflict of interest statement in the “Confidential to Editor” section, and submit your "Accept" recommendation.

Reviewer #1: All comments have been addressed

Editor: OK

2. Is the manuscript technically sound, and do the data support the conclusions?

Reviewer #1: Yes

Editor: Yes

3. Has the statistical analysis been performed appropriately and rigorously? 

Reviewer #1: Yes

Editor: Yes

4. Have the authors made all data underlying the findings in their manuscript fully available?

Reviewer #1: No

Editor: Yes (as declared)

5. Is the manuscript presented in an intelligible fashion and written in standard English?

Reviewer #1: Yes

Editor: Yes

6. Review Comments to the Author

Reviewer #1: The manuscript was well-written. The authors addresses the factor from the introduction to the discussion. However, I was wondering if the media exposure is related to the level of education of the mothers. Also if the mothers that are advanced in age and are highly educated with good media exposure are still at risk of maternal mortality. What will be the implication to the government policy.

Editor: Please see in file attached a minor error requiring correction.

7. PLOS authors have the option to publish the peer review history of their article (what does this mean?). If published, this will include your full peer review and any attached files.

Reviewer #1: No

---

## [Editor Report · Acceptance letter]

12 Nov 2024

PONE-D-24-24034R1 

PLOS ONE

Dear Dr. Bezie, 

I'm pleased to inform you that your manuscript has been deemed suitable for publication in PLOS ONE. Congratulations! Your manuscript is now being handed over to our production team.

Kind regards, 

on behalf of

Dr. Alfredo Luis Fort 

Academic Editor

PLOS ONE